# Alternative Methods to Current In Vivo Procedures to Address the 3Rs Tenet in Rabies Proficiency Testing

**DOI:** 10.3390/v14081698

**Published:** 2022-07-31

**Authors:** Maira Zorzan, Morgane Gourlaouen, Stefania Leopardi, Paola De Benedictis

**Affiliations:** Food and Agriculture Organization of the United Nations, Italian Reference Centre for Rabies, Istituto Zooprofilattico Sperimentale delle Venezie, viale dell’Università 10, 35020 Legnaro, Italy; morgane.gourlaouen@fao.org (M.G.); sleopardi@izsvenezie.it (S.L.); pdebenedictis@izsvenezie.it (P.D.B.)

**Keywords:** rabies, proficiency testing, 3Rs tenet, in vitro/ex vivo technologies

## Abstract

Canine rabies is responsible for an estimated 59,000 human deaths every year. In an attempt to reach the ZeroBy30 goal, robust disease surveillance coupled with improved diagnostics play a paramount role in ensuring reliable data and gradually attesting rabies control advancements. In this context, proficiency testing is organized to harmonize rabies diagnostic capacities. In most exercises, rabies-positive samples consist of brains collected from intracerebrally inoculated mice. This procedure causes distress and severe suffering to animals, raising important ethical concerns that can no longer be ignored. In the last decades, the 3Rs tenet (Replace, Reduce, Refine) has been successfully implemented in several scientific areas, and we strongly support its application in the framework of rabies proficiency testing. Here, we discuss cell-based technologies as innovative sustainable in vitro candidate systems to replace in vivo experiments for the production of proficiency testing samples. The application of these alternative methods can allow completely in vitro or ex vivo production of rabies proficiency testing panels, which would represent an important replacement or reduction/refinement for current in vivo procedures.

## 1. Introduction

Rabies is a zoonotic infectious disease causing an estimated 59,000 human deaths every year, mainly in rural areas of the African and Asian continents, with the vast majority of human deaths mediated by dogs [1,2,3]. In endemic areas, weak surveillance data of both human and animal cases have fostered the cycle of neglect, leading to very few resources mobilized by stakeholders and, consequently, to poor prevention and control measures. For this reason, robust disease surveillance supported by improved diagnostics have been unanimously fixed as an indicator of Objective 2.2 (Ensuring reliable data for effective decision making) under the ZeroBy30 Global Strategic Plan to end human deaths from dog-mediated rabies [4,5].

In this context, proficiency testing (PT) for rabies diagnosis is a substantial tool to harmonize the diagnostic capacity of participating laboratories, providing in turn guidance for the improvement of diagnostic skills [3]. An interlaboratory exercise (although slightly different, in the present manuscript PT and interlaboratory exercise are used synonymously) involves the PT provider, acting as a reference for rabies diagnosis, and a variable number of diagnostic laboratories receiving a panel of blind-coded samples to be tested by rabies diagnostic techniques routinely used for field samples [6]. Collected data are analyzed by the PT provider that reports on each participant’s performance, highlighting the laboratory diagnostic capability and suggesting practical recommendations for future improvement, if needed. Particularly in resource-limited countries, the PT providers should also offer its support to diagnostic laboratories with respect to technical training of lab staff, supplying protocols and reagents, as well as setting up and validating new techniques [7]. PT panels for the diagnosis of animal rabies usually include samples that are either negative or positive for the presence of lyssavirus, including the virus, antigen and/or viral RNA, according to the diagnostic techniques to be applied [8]. Samples may consist of either brain tissue impression slides [9] or lyophilized brain material [8,10]. Positive samples mostly include different strains of rabies virus (RABV) and possibly non-RABV lyssavirus species that, although rare, are included in the PT panels to evaluate the breadth of diagnostic capabilities. Ideally, the use of field samples would allow preparing PT panels that are representative of naturally occurring animal rabies cases. However, rabies-positive field samples are rarely available in large amounts, and the required panel homogeneity may not be ensured if this kind of material is used [9]. For this reason, in most PT exercises, positive samples are produced by the intracerebral inoculation of mice [8,10]. Approximately five hundred mice are needed for a PT exercise involving about thirty laboratories, and interlaboratory trials are regularly organized, as labs should ideally participate once every one or two years. Although this practice is still widely adopted, it is now time to ask ourselves whether, with respect to animal welfare, more acceptable procedures could be developed, particularly if we consider the strong pressure to dismiss the mouse inoculation test (MIT) in favor of more recent techniques, such as the cell culture test [11]. Apart from being a painful and invasive procedure, the intracerebral injection of live rabies inflicts severe suffering to animals. In order to secure a great amount of the virus in brain tissue, the experimental endpoint of this technique lies in the full display of neurological symptoms and coincides with the human endpoint.

Over sixty years have passed since William Russell and Rex Burch launched the 3Rs tenet (Replace animals with non-sentient alternatives, Reduce the number of animal used, Refine experiments to improve animal welfare and reduce animal distress), an initiative that has gained increasing attention, positively influencing international guidelines and local governmental legislation, as well as the overall approach to animal experimentation by scientists worldwide [12]. The great advances observed in recent years regarding the production and maintenance of several cellular systems and cell-based substrates, strongly suggest that the time is ripe to make a step forward in the framework of rabies proficiency testing and to develop an innovative sustainable in vitro system to replace or at least reduce and/or refine current in vivo experiments. In the present article, we sift through all alternative options we believe may make the difference and revolutionize the preparation of PT panels for rabies diagnosis and positive controls for routine diagnosis.

## 2. Hypothesis and Significance

We strongly believe that cell-based systems represent a great opportunity to replace the use of animals in the framework of rabies proficiency testing. Traditional cell cultures growing either in suspension or as cell monolayers, ex vivo organotypic brain cultures and more modern 3D cellular models all represent sustainable substrates that can be exploited for the in vitro amplification of rabies virus and the collection of infected material, thus avoiding or significantly reducing the use and suffering of animals (see Section 4 for more details). Currently used brains from experimentally infected mice can indeed be replaced by cellular material either produced in a totally animal-free way or collected from healthy donors and can be subsequently in vitro infected. This latter technique would represent an important refinement compared to current animal experimentation, as animals would be euthanized to collect brains for subsequent in vitro infection but would not suffer from a rabies infection. Therefore, the application of in vitro technologies in rabies diagnostic proficiency exercises can potentially enable the implementation of the 3Rs tenet (Replacement, Reduction and Refinement) that is strongly sought in modern science. In the field of rabies diagnostics and anti-rabies vaccine production, the application of in vitro techniques has greatly contributed to significant reductions in animal use for rabies virus isolation and the determination of vaccine potency, as detailed in the following section (see Section 3). Altogether, these examples further support that a change in the course in the preparation of PT panels is also possible (Figure 1).

## 3. Examples from the (Recent) Past

Rabies virus isolation by the MIT represented a common practice for diagnostic purposes worldwide for decades, until the advent of cellular cultures and the development of the rapid tissue culture infection test (RTCIT), which led to a gradual but substantial change in the routine diagnostics of several laboratories [6,11]. Besides the ethical issue associated with the MIT, such a technique is more expensive, laborious, time-consuming and requires much more time (at least 21 days) for reliable results compared to RTCIT [10]. Conversely, the isolation of rabies viruses in cell cultures typically gives results in 3–9 days and can be performed in laboratories with standard equipment and no animal facilities. Moreover, the introduction of RTCIT allowed not only the avoidance of mouse suffering but also a complete replacement of animal use for rabies diagnostics. For these reasons, the application of RTCIT instead of MIT is, today, strongly recommended by all international organizations [6,11]. Nevertheless, RTCIT still presents some limitations, especially in low-resource settings, where dedicated cell culture facilities as well as biosafety requirements are not always sustainably available [13]. In this context, the implementation and general recognition of molecular methods, such as routine rabies diagnostic tests, have the potential to overcome the use of both MIT and RTCIT, which are both less sensitive, as they need a viable virus, meaning fresh or correctly stored samples.

Another important implementation of the 3Rs tenet was introduced some years ago in the framework of vaccine potency testing. The National Institutes of Health (NIH) test requiring the intracerebral virus challenge of mice after immunization was traditionally used to evaluate the potency of human and veterinary rabies vaccines before batch release [14]. However, the NIH test raises several questions regarding the high variability of results, costly biosafety requirements and, of course, the ethics of such a severe animal procedure [15]. International organizations and expert working groups have largely encouraged replacing the NIH test with in vitro methods; this has led to the development of several alternative assays, including the antibody binding test [16], the single radial immunodiffusion test [17] and some enzyme-linked immunosorbent assay (ELISA) methods to measure the vaccine content of rabies glycoprotein, the major protein stimulating the host’s immune system [18,19,20]. The European Pharmacopoeia currently accepts a validated immunochemical or serological potency assay to evaluate human vaccine potency, [21] and this represents a great example of 3Rs implementation, in particular as an important replacement of animal use. Concerning rabies vaccines for veterinary use, many research studies reported difficulties in the development of ELISA assays able to reliably determine vaccine potency due to significant interference by vaccine adjuvants [18]. However, a serological potency assay has been validated for rabies vaccine for veterinary use in a collaborative study organized by the European Directorate for the Quality of Medicines and HealthCare (EDQM) and has been recommended to experts of the European Pharmacopoeia [22]. By avoiding the virus challenge and by simply requiring the antibody titration of immunized mice, this serological assay represents a significant refinement compared to the classical NIH test.

## 4. Candidate Cell-Based Technologies and Risk Analysis

Several in vitro cellular systems have largely been used in human and veterinary medicine to replace in vivo experiments for research and diagnostic purposes. Most of these techniques may find applications in replacing mice used in the preparation of sample panels for rabies interlaboratory trials. In order to prepare rabies-positive samples, brains collected from intracerebrally infected mice are often diluted in cerebral material collected either from large mammals available at the laboratory, large animal facilities or from field carcasses previously tested negative for rabies and for a variety of neurotropic pathogens. With the aim of an innovative implementation of the 3Rs tenet, a cell substrate infected with the rabies virus should therefore replace infected brains obtained through laboratory mice infection.

The most simple and attractive method would be an in vitro infection of cell monolayers followed by the collection of infected cells mimicking the infected mouse brain. Several cell lines, including neuroblastoma, glioma and fibroblast cell cultures, are sensitive to the rabies virus infection and are frequently used for the cell adaptation and amplification of different rabies strains. Thus, each of these cell lines or a mixture of them could potentially replace mouse brains to produce rabies-positive samples. In addition, some cell lines can be adapted to suspension cultures, either as single cells or spheres, making it easier to scale up to the needed volumes for the production of a high number of samples. Clearly, cell lines may not be equally suitable for the replication and massive production of non-RABV lyssaviruses, a drawback deserving further research and development efforts. Regardless, cell culture conditions, including for instance multiplicity of infection, time of incubation and selection of proper culture media, should be experimentally determined in order to maximize the collection of infected cells. One of the main concerns from replacing infected brain tissues with infected cell lines is that they may differ in the expression pattern of viral antigens detectable through the gold standard techniques, the Direct Fluorescent Antibody (DFA) test or the Direct Rapid Immunohistochemistry Test (DRIT) [6], leading to PT samples far away from the field ones. We believe that co-cultures of different cell types or the mixing of different cell lines could more consistently mimic the features of a brain of an infected animal. Regardless, innovative in vitro samples should pass a blind test before being considered for further development in this topic. Preliminary attempts for using infected cell lines have proved to be promising in replacing standard in vivo experiments. Indeed, the fluorescent pattern of the alternative in vitro method is substantially undistinguishable from that obtained by the standard one (Figure 2).

Other options are available to replace animals in case non-primary cell lines do not prove suitable. Three-dimensional brain models, such as cerebral organoids and organotypic brain slice cultures, should also be evaluated as substitutes for mouse brains. These systems have been successfully used for research on brain pathophysiology [23,24], the modeling of viral infections [25,26] and the study of oncolytic virus therapies [27], suggesting that they could also be adapted to the in vitro amplification of the rabies virus. In particular, the set up and in vitro infection of cultures of brain explants from healthy large mammals could potentially represent an easy way to collect a relatively high volume of infected cellular material, with a great experimental refinement of animal welfare compared to current painful procedures. However, particular attention should be posed towards homogenizing the material prior to sample preparation, as different brain portions may be differently permissive to viral infection and/or show differential expression patterns of viral antigens. The application of cerebral organoids in the framework of rabies proficiency testing is certainly more ambitious and complex, as the establishment of these cultures may be long and require the testing and setting up of a high number of variables. Nevertheless, the presence of different cell types within the organoids and the obtained cellular mass could make the difference in the definition of the most suitable cellular system.

## 5. Assessing Molecular Diagnostic Capabilities

Molecular methods, including both end-point and real-time RT–PCR, have been recently mentioned among the reliable diagnostic methods [11] to confirm results obtained by the DFA test or in the case of poor quality samples (i.e., putrefied field carcasses) giving inconclusive results by DFA assay. Some proficiency tests may therefore be specifically aimed at assessing the molecular diagnostic capabilities of participating laboratories. Of note, this type of PT exercise offers the possibility to implement the 3Rs tenet much more easily than the one focused on DFA and virus isolation and is worth further discussion in the context of the present topic. The first interlaboratory trial for the detection of lyssavirus RNA through RT–PCR was organized by Friedrich-Loeffler-Institut (FLI) and involved 16 European laboratories analyzing a panel of 26 RNA samples previously extracted either from original brain material or cell culture supernatant [28]. No animal infections were actually performed for the preparation of these panels, and this is of course a noteworthy achievement in the framework of the 3Rs tenet. However, the extraction of viral RNA from the diagnostic sample may represent a critical step in the diagnostic process and should be performed at the laboratories participating in the PT exercise for a more reliable evaluation of their performances. This is, in fact, as important as obtaining samples of proper quality for techniques such as DFA and DRIT. In this context, a step forward has been represented by preparing PT samples consisting of uninfected cerebral material spiked with viral RNA, as recently described by our group [8]. To produce these samples, testing negative by the DFA test and positive by RT–PCR, no mice were experimentally infected, and RNA extraction was still part of the diagnostic process performed at the participating laboratories. The viral RNA may be synthetic or total, previously extracted from field samples or cell culture supernatants. It is very important to note that RNA-spiked samples prove very stable despite the known instability of RNA and also present the benefit of being completely non-infectious. The opportunity to prepare non-infectious PT panels is a great achievement with respect to biosafety for both carriers and lab staff and allows a significant reduction in delivery costs. Finally, the molecular PT exercises here reported demonstrate that a partial yet important implementation of the 3Rs tenet has already been validated, whereas more research studies are needed to develop a completely in vitro or ex vivo production of rabies PT panels.

## 6. Conclusions

The 3Rs tenet is one of the most sponsored approaches in worldwide animal sciences. This is not only for indubitable ethical reasons but is also for the impact of the use of a large number of animals overall in the environmental chain. In agreement with this consideration, a reduction in the number of animals is associated with a reduction in costs and decreased consumption of resources towards more sustainable science. The validation of an in vitro or ex vivo model to replace or reduce/refine the use of animals in interlaboratory trials in rabies diagnosis represents an intriguing challenge. In addition, a deep investigation of all possible applications of classical and more recent cellular models to rabies virus research may have an important impact not only on rabies proficiency testing but also on pathogenicity and therapeutic studies. In conclusion, we believe that the implementation of the 3Rs tenet in the framework of rabies interlaboratory trials is worth a deep investigation, as modern available cell-based technologies can make an impactful difference.

## Figures and Tables

**Figure 1 viruses-14-01698-f001:**
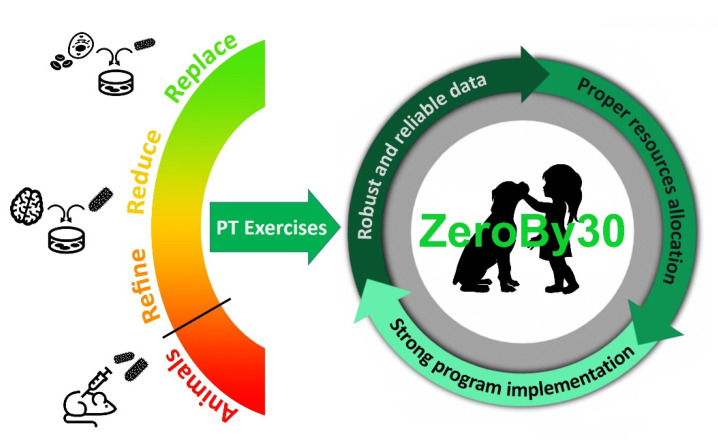
In vitro and ex vivo methods can potentially enable the implementation of the 3Rs tenet (Replacement, Reduction, Refinement) in the framework of rabies PT exercises, which in turn contribute to guaranteeing robust and reliable diagnostic data towards the ZeroBy30 global goal.

**Figure 2 viruses-14-01698-f002:**
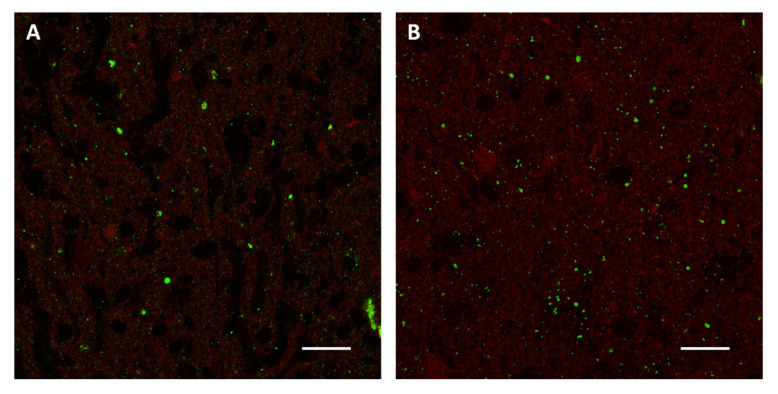
Direct Fluorescent Antibody test performed on samples prepared using the standard in vivo method (**A**) and an alternative in vitro method based on infected BSR and Neuro-2a cells (**B**). An anti-rabies nucleocapsid conjugate (Biorad) was used for the detection of rabies virus antigens (green). Images were acquired by Leica SP8 confocal microscope (63× objective, 25 µm bar).

## Data Availability

All data are contained within the present article.

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
