# Peer review of "Alternative Methods to Current In Vivo Procedures to Address the 3Rs Tenet in Rabies Proficiency Testing"

_viruses, 2022, doi:10.3390/v14081698_

Round 1

Reviewer 1 Report

The paper titled “Seroconversion Status and Ensuring reliable rabies surveillance data whilst addressing the 3Rs tenet” by Zorzan et al. presents both a good overview of the need for rabies diagnostic proficiency testing for the ZeroBy30 goal and promoting in vitro means in providing the necessary samples. The authors covered the challenges in design and garnering the material for rabies proficiency testing very well and can serve as an informative guide on this topic. The argument for implementing the 3Rs approach is strong, clear, and convincing, especially at this moment of global effort to control and prevent human rabies deaths from dog bites and of recent publications on novel in vitro cell culture and ex-vivo biological models for brain disease research. While this paper does a very good job of proposing valid replacements for mouse infected brains as source for the proficiency samples, a downside was not covered, such as will eliminating the local microenvironment of an infected brain by these ex-vivo models of organs also eliminate the source of potential non-specific reactions in DFA and DRIT (considering ‘clean’ lab sourced samples vs. field samples). Ideally, the proficiency sample will represent the samples submitted for analysis, but even infected mouse brains are not ideal for this purpose. A suggested improvement is to mention/review what a proficiency test can do and cannot do regarding purpose. For example, for the purpose of evaluating the laboratory’s ability to detect in brain cells the presence or absence of rabies virus/antigen/RNA, not considering methods in obtaining the sample, the argument for cell culture and ex-vivo models is very strong. However, these samples cannot mimic exactly (as currently described) field samples, thus cannot evaluate the steps of the assay for sample preparation and potentially sources of non-specific reactions.

Below are very minor comments for improvement:

Line 15 – change ….”testing are organized’ to “testing is organized”

Line 149 – is the word “paper” correct, not sure what word was intended?

Line 210-211 – addressed the importance of obtaining the sample properly in context of viral extraction for molecular methods, it should also be mentioned that obtaining the proper sample for DFA and DRIT is also important.

Line 264 – please check the publication year.

Reviewer 2 Report

In this manuscript by Zorzan et al., the importance of looking at the methods used for proficiency testing is discussed. Overall, the manuscript is well written and ideas are clearly communicated while providing some interesting topics for future research. As such, I only have a few minor comments.

1. I find the title a bit misleading. While participating in PT schemes are important to indicate diagnostic capacity and is a integral part for quality management it is only an indirect part of ensuring reliable surveillance data. The title should rather reflect the main message of the manuscript i.e. alternative methods to current in vivo procedures.

2. Section 2, lines 82-99. The authors should please include references for the specific techniques mentioned or refer to other sections where these techniques are discussed in more detail.

3. Line 88-92. The authors state that in stead of using brain material from experimentally infected mice, the use of healthy donors should be considered. Could the authors please clarify what is meant by this? How would this work? Do they mean euthanising healthy animals to remove brain material for in vitro infection, since this will not reduce the number of animals needed?

4. Lines 110-114. While I agree that RTCIT is a better technique than MIT ethically it is not always feasible especially in low resource settings. Dedicated cell culture facilities will be required (depending on your definition of standard lab equipment). Depending on the specific lyssavirus species different biosafety considerations might be applicable. So while RTCIT is an improvement over MIT, it still has its limitations and there is room for new/better techniques
